# Effects of Concentrate Supplementation on Growth Performance, Rumen Fermentation, and Bacterial Community Composition in Grazing Yaks during the Warm Season

**DOI:** 10.3390/ani12111398

**Published:** 2022-05-29

**Authors:** Dongwen Dai, Kaiyue Pang, Shujie Liu, Xun Wang, Yingkui Yang, Shatuo Chai, Shuxiang Wang

**Affiliations:** Key Laboratory of Plateau Grazing Animal Nutrition and Feed Science of Qinghai Province, Academy of Animal Science and Veterinary Medicine, Qinghai University, Xining 810016, China; daidongwen@stu.nxu.edu.cn (D.D.); pky0425@163.com (K.P.); mkylshj@126.com (S.L.); wangxun513@163.com (X.W.); yangyikui721@163.com (Y.Y.)

**Keywords:** yak, growth performance, serum, rumen fermentation, rumen bacterial, concentrate supplementation

## Abstract

**Simple Summary:**

Yaks are an important for the economy and livelihood of local herders in the Tibetan Plateau and are their main source of income. They are mainly fed on natural pastures, especially during the warm season. Information on the effects of concentrate supplementation on the growth performance, rumen fermentation, and bacterial community composition of grazing yaks during the warm season is limited. Our study showed that concentrate supplementation increased the relative abundances of Firmicutes, *Succiniclasticum*, *Prevotellaceae_UCG_003*, *Prevotellaceae_UCG_005*, and *Ruminococcus_1*. Supplementation increased the average daily gain (ADG) and serum concentrations of glucose (GLU), total protein (TP), and aspartate aminotransferase (AST). Furthermore, the concentrations of ruminal ammonia-N (NH_3_-N), microbial protein (MCP), and volatile fatty acids (VFAs) increased in the supplement group. We concluded that supplementary feed improved ruminal fermentation, and altered the bacterial community composition in yaks during the warm season, thereby improving growth performance.

**Abstract:**

This study aimed to evaluate the effects of concentrate supplementation on the growth performance, serum biochemical parameters, rumen fermentation, and bacterial community composition of grazing yaks during the warm season. Eight male yaks (body weight, 123.96 ± 7.43 kg; 3-years) were randomly allocated to two treatments groups: grazing (n = 4, GY) and concentrate supplement group (n = 4, GYS). Concentrate supplementation increased the average daily gain (ADG) *(p* < 0.05). Glucose (GLU), total protein (TP), and aspartate aminotransferase (AST) serum concentrations were significantly higher in the GYS group than in the GY group (*p* < 0.05). Ammonia-N, MCP: microbial protein, and total volatile fatty acid concentrations were significantly higher in the GYS group than in the GY group (*p* < 0.01), whereas the pH and acetate: propionate values were significantly decreased (*p* < 0.01). The relative abundance of Firmicutes in the rumen fluid was significantly higher in the GYS group than in the GY group (*p* < 0.01). At the genus level, the relative abundances of *Succiniclasticum*, *Prevotellaceae_UCG_003*, *Prevotellaceae_UCG_005*, and *Ruminococcus_1* were significantly greater in the GY group than in the GYS group (*p* < 0.01). In conclusion, concentrate supplementation improved yaks’ growth potential during the warm season, improved ruminal fermentation, and altered core bacteria abundance.

## 1. Introduction

Yak (*Bos grunniens*) mainly inhabits the Qinghai–Tibet Plateau at an altitude range from 2500–5500 m, adapting well to harsh climatic conditions and accounting for approximately 90% of the total yak population worldwide [1]. These animals provide meat, milk, wool, and fuel for local nomads and are an important source for local economy and livelihood [2,3]. In traditional management, yaks usually graze on a full grazing system with only herbage as feed. However, the nutrition of forage grasses undergoes a dynamic change; normally, there are two main periods, that from May to September (warm season) with increased in forage supply and that from October to April (cold season) with reduced in forage supply [4]. Therefore, supplementary feeding measures for yaks are mainly performed in the cold season, and it has been revealed that supplemental feeding can improve growth performance [5,6]. However, it remains unclear whether supplemental feeding of yaks in the warm season improves growth performance and boosts economic efficiency.

The rumen of ruminants is equivalent to an anaerobic fermentation tank, mainly inhabited by microorganisms such as bacteria, fungi, and protozoa, which can degrade the feed entering the rumen [7]. Ruminants rely on rumen microorganisms to degrade dietary fiber, starch, proteins, and other nutrients and synthesise volatile fatty acids (VFAs) and microbial proteins, which play an essential role in nutrient metabolism [8]. Volatile fatty acids are the primary energy sources of the living host, providing 70–80% of the energy required [9]. Therefore, ruminal microbes are essential for ruminant nutrient metabolism and absorption, immune responses, and gastrointestinal development [10]. Previous studies have shown that species, diet, age, and season affect rumen microorganisms, with diet being a crucial factor [11,12]. Therefore, it is imperative to fully understand the response of yak rumen microorganisms to changes in the diet to improve yak growth performance.

There have been several studies on yak rumen microbes [13,14], and the response of yak rumen microbes to diet structure has become a hot research topic in recent years [15,16]. For instance, Liu et al. [17] investigated the effects of supplementation with concentrates or per-rumen lysine and methionine on rumen microbes in lactating yaks. Similarly, current research on yak supplementation has focused on the cold season [18,19]. However, to the best of our knowledge, few studies have investigated the effects of concentrate feed supplementation on rumen fermentation parameters and the bacterial composition of grazing yaks during the warm season. Therefore, our research focused on this area.

We hypothesised that concentrate supplements would affect rumen fermentation, microbial structure and function, and organismal metabolism, enhancing growth performance. Hence, our study aimed to comprehensively investigate the effects of concentrate supplementation on rumen fermentation and the microbiota of yaks during the warm season and provide a theoretical basis for supplementary feeding of yaks. We used 16 S rRNA sequencing technology to study the yak rumen microbiota community and discussed the possible relationship between fermentation parameters and microorganisms.

## 2. Materials and Methods

All experimental procedures and animal experiments were performed in accordance with the guidelines of the Ethics Committee. This study was approved by the Institutional Animal Care and Use Committee of the Qinghai University (Protocol number: QHU20190812).

### 2.1. Study Site

This study was conducted from July to September 2019 at a Laozaxi pasture (35°3411″ N, 100°4645″ E, altitude 3100 m above sea level) in Guinan County, Qinghai Province. The dominant species of the forage were *Elymus nutansv*, *Kobresia humili*, *Poa pratensiv*, *Potentill abifurca*, *Saussurea pulchra*, and *Ajania tenuifolia* during the experimental period. The average temperature during the study period was 7 °C and the mean precipitation was 116.34 mm, and these data were obtained from Qinghai meteorological station.

### 2.2. Animals, Diet, Experimental Design, and Management

Eight 3-year-old male yaks with similar weight (123.96 ± 7.43 kg) were randomly divided into two groups. One group was only grazed (GY), and the other group was grazed + concentrate supplemented (44.27% corn, 12.06% wheat bran, 12.88% rapeseed meal, 4.98% wheat, 12.28% soybean meal, 8.57% cottonseed meal, 1.22% Ca(HCO_3_)_2_, 0.87% NaHCO_3_, 0.87% NaCl, 2.00% Premix; GYS), separated by fencing. Yaks were herded in two enclosure areas of the same size in the same pasture, and they were free to drink water during the experiment. Grazing started at 07:00 and ended at 18:00. A concentrate supplement was provided at 18:00. Yaks in the supplementation group received 1% of daily concentrate supplementation per body weight of yak. The forage and concentrate nutrient levels are listed in Table 1. The pre-experimental period lasted for 10 d, and the experimental period lasted for 60 d.

Before morning feeding, yak weight was measured continuously at the beginning and end of the trial using floor scales, and daily weight gain (ADG, g/d) was calculated as = (Initial BW−Final BW)/60.

### 2.3. Sample Collection

Mixed forage samples were retrieved from a pasture grazed on by yaks. Quadrats of 0.5 × 0.5 m were randomly placed across the pasture to mimic the yak selection of forage when grazing. Inedible forage was removed when collecting mixed forage samples, and only edible forage was retained. They were then stored in self-sealing bags at −20 °C for later analysis.

After the experiment (70 days), blood samples were collected from the jugular vein into 5 mL evacuated tubes without anticoagulant before morning feeding. Samples were centrifuged (4000× *g*, 15 min, 4 °C), frozen with liquid nitrogen, and transferred to a −80 °C refrigerator. Approximately 150 mL of rumen fluid samples were collected from each animal using a stomach tube before morning feeding, as previously described by Bunemann et al. [20]. T The tube was thoroughly cleaned between sample collections, and the first 50 mL of liquid was discarded to minimise saliva pollution. The collected rumen fluid was filtered through four layers of sterile gauze, and rumen pH was measured by portable pH on-site (Testo205; Testo AG, Schwarzwald, Germany). The rumen fluid was divided into two parts to analyse rumen fermentation parameters and extraction of DNA, then frozen with liquid nitrogen and transferred to a −80 °C refrigerator.

### 2.4. Feed Analysis

Forage and concentrate feed samples were dried in an oven at 60 °C for 48 h until they had constant weights. Then, they were ground in a small hammer mill and passed through a 1 mm sieve (40 mesh) before determining the chemical composition. We measured the composition (dry matter, crude protein and fat, potassium, and calcium) of the concentrate and forage-based concentrate using an AOAC procedure (1990) [21]. The neutral detergent fibre (NDF) and acid detergent fibre (ADF) contents were determined using methods described by Van Soest et al. [22].

### 2.5. Serum and Rumen Fermentation Parameters

The serum concentrations of glucose (GLU), triglycerides (TG), total protein (TP), urea nitrogen (UN), aspartate aminotransferase (AST), and alkaline phosphatase (ALP) were analysed using a chemical analyser (Beckman Coulter AU 480; Brea, CA, USA).

For determination of VFAs (acetic, propionic, isobutyric, butyric, isovaleric, and vleric acids), a 2 mL sample of thawed ruminal fluid was placed in a centrifuge tube, mixed uniformly with 0.5 mL of 25% ortho-phosphoric acid, and clarified by centrifugation at 13,000× *g* rpm at 4 °C for 15 min before analysis by gas chromatography (7980 A; Agilent Technologies, USA). The determination conditions were as follows: column temperature (FFAP 30 m × 0.32 mm × 0.5 um) 110 °C, inlet temperature 220 °C, detector temperature 220 °C, splitting ratio 30:1, high purity nitrogen as the carrier gas, as described by Zhang et al. [23]. Concentrations of ammonia nitrogen (NH3-N) were determined according to a phenol hypochlorite assay, as described by Broderick et al. [24], and ruminal microbial protein (MCP) was determined based on the method described by Makkar et al. [25]. Microbial proteins were estimated from the purine to N ratio in the isolated bacteria. Yeast RNA was used as the standard.

### 2.6. DNA Extraction and Determination

Total gDNA was extracted from ruminal fluid samples using the E. Z. N. A ^®^kit (Omega Bio Tek, Norcross, GA, USA) according to the manufacturer’s instructions. The concentration and purity of DNA were determined using NanoDrop 2000 c (Thermo Scientific, Wilmington, DE, USA), and the integrity of DNA was evaluated by 1% agarose gel electrophoresis PCR amplification was performed using primers for DNA located on both sides of the V3 + V4 high variant region, with the standard primers 338 F: (5′-ACTCCTACGGGAGGCAGCA-3′) and 806 R (5′-GGACTACHVGGGTWTCTAAT-3′). The PCR reaction system used was as follows: 5 μL KOD FX Neo Buffer, 2 μL dNTP (2 mmol/L), 0.3 μL primer (10 μmol/L), 50 ng no of template DNA, and 0.2 μL of KOD ROD polymerase. The reaction parameters were as follows: predenaturation at 95 °C for 5 min, followed by denaturation at 95 °C for 30 s, annealing at 50 °C for 30 s, extension at 72 °C for 1 min for a total of 25 cycles, and final extension at 72 °C for 7 min. Magnetic DNA beads (0.8 XM) purified the amplified products, and the purified DNA fragments were recovered by 1.8% agarose gel cutting. DNA fragments were sequenced using the Illumina HiSeq 2500 sequencing platform at *Beijing Baemai Biotechnology Co. Ltd.* (Beijing, China) to obtain raw data. The following sequencing parameters were applied: paired-end sequencing run type, 2 × 300 bp read length. We obtained 1,068,537 raw reads, which were extracted into FASTQ format and used for subsequent bioinformatic analysis. FASTQ files were deposited in the Sequence Read Archive of NCBI (accession no. PRJNA635784).

### 2.7. Data and Statistical Analysis

The original data were spliced and filtered using Flash (V.1.2.11) and Trimmomatic (0.36), chimeras were removed using UCHIME (8.1), and a series of high-quality samples were obtained. Using UPARSE (V7.0.1090), OTU delineation of sequences, and bioinformatics statistical analysis was based on a 97% similarity level [26].

Comparison of Ribosome Database items was performed with a (RDP) classifier (v2.11) with the SILV A (SSU123) database 2, with a comparison threshold of 70%, and each sequence was annotated. Alpha diversity was analysed using two indicators, Chao1 and Shannon index, and was calculated using MOTHUR (versionv.1.30.1). Beta diversity was estimated by calculating the unweighted UniFrac distances and visualised using principal coordinate analysis (PCoA). PICRUSt 2 was used to predict the metabolic pathways of the microbial communities in samples from different groups (based on KEGG).

Statistical analyses were performed using SPSS 24.0 (SPSS Inc., Chicago, IL, USA). The statistical difference between normally distributed data was analysed using an independent sample *t*-test. The alpha diversity, bacterial composition, and metabolic pathway abundances of function prediction were non-normal and were analysed using Kruskal–Wallis tests. Spearman correlation analysis was used to analyse the correlation between fermentation parameters and predominant genera, and *p*-values ˂ 0.05 indicated significant differences.

## 3. Results

### 3.1. Growth Performance

The effects of concentrate supplementation on the growth performance of the grazing yaks are shown in Table 2. The initial body weight was not different between GY and GYS groups, but after 60 days, the GYS group had a greater body weight than the GY group (*p* < 0.05), and the ADG was 170.27% higher in the GYS group than in the GY group (*p* < 0.05).

### 3.2. Serum Biochemical Parameters

The serum biochemical parameters of the grazing yaks in the GY and GYS groups are shown in Table 2. Serum concentrations of GLU, TP, and AST were higher in the GYS group than in the GY group (*p* < 0.05). 

### 3.3. Rumen Fermentation Parameters

NH_3_-N concentration, VFA production, and proportions of acetate, propionate, isobutyrate, butyrate, isovalerate, and valerate were significantly higher in the GYS group than in the GY group (*p* < 0.05; Table 3). However, the pH value and acetate: propionate ratio were significantly decreased when yaks were supplemented with different concentrations (*p* < 0.05).

### 3.4. Alpha Diversity and Composition of Rumen Bacterial Communities

A total of 959,181 effective 16 S rDNA gene sequences were obtained from eight rumen fluid samples. After subsampling each sample to an equal sequencing depth (29,114 reads per sample) and clustering, we obtained 2055 OTUs with a recognition rate of 97%. The 1810 OTUs were shared, accounting for 88.1% of the total number of OTUs. The number of unique OTUs in GY and GYS was 131 and 113, respectively (Figure 1C). The Shannon index indicates microbial diversity, whereas the Chao1 value shows its abundance, with higher values indicating higher bacterial community diversity and abundance. The Chao1 value (1905.82 ± 48.90 vs. 1718.91 ± 33.98, *p* < 0.05) (Figure 1B) and Shannon index (8.54 ± 0.15 vs. 8.01± 0.32, *p* < 0.05) (Figure 1A), revealed that the diversity and abundance of ruminal fluid microorganisms in GYS group were significantly higher than that of the GY group. 

We identified 24 bacterial phyla in samples from different feed types based on taxonomic analysis. The dominant phyla were Firmicutes, Bacteroidetes, Saccharibacteria, and Fibrobacteres. Firmicutes and Bacteroidetes were the most representative phyla among all samples, accounting for 92.76% and 94.30% of the total reads, respectively (Figure 2A). At the genus level, 199 genera were detected in the yak rumen samples. The predominant genera were *Prevotella 1* (8.99%), *Christensenellaceae_R-7_group* (13.23%), *Rikenellaceae_RC9_gut_group* (6.63%), *Succiniclasticum* (3.49%), *Eubacteriumcoprostanoligenes_group* (3.87%), *Prevotellaceae_NK3B31_group* (1.76%), *Ruminococcaceae_NK4A214_group* (7.74 %), and *Prevotellaceae_UCG-003* (1.86%), respectively (Figure 2C).

### 3.5. Differences in Rumen Bacterial Community Composition

At the phylum level (Figure 2B), the relative abundance of Firmicutes was significantly higher in the GYS group than in the GY group (*p* < 0.01), whereas the relative abundance of Bacteroidetes was significantly higher in the GY group than in the GYS group (*p* < 0.01). At the genus level (Figure 2D), the relative abundances of Succiniclasticum, Prevotellaceae_UCG_003, Prevotellaceae_UCG_005, and Ruminococcus_1 were higher in the GY group than in the GYS group (*p* < 0.05). In contrast, the relative abundances of *Christensenellaceae_R-7_group* and *Ruminococcaceae_NK4A214_group* were higher in the GYS group than in the GY group (*p* < 0.05). Beta diversity was assessed using principal coordinate analysis. PCoA of the unweighted UniFrac distance matrices (Figure 1D) revealed differences in the bacterial community structures in the samples. The study showed that the bacterial community structure was significantly different between the GY and GYS groups, indicating that supplementary feeding affected the composition of rumen community bacteria of grazing yaks during the warm season.

### 3.6. Correlation between Rumen Fermentation Parameters and Bacteria Community

Based on Spearman’s correlation coefficients, we analysed the correlation of the main ruminal microbiota (at the genus level) with rumen fermentation parameters (Figure 3). The pH value of the ruminal fluid was positively correlated with the relative abundance of the *Prevotellaceae_NK3B31_group* and *Prevotellaceae_UGG-003*. The molar proportion of acetate was positively correlated with the *Prevotellaceae_NK3B31_group* and *Prevotellaceae_UGG-003*. The MCP concentration was positively correlated with the *Prevotellaceae_NK4A214_group* and negatively associated with *Ruminococcus_1*. TVFA concentration was positively correlated with *Prevotellaceae_NK4A214_group* and negatively associated with *Ruminococcus_1*. The molar proportion of butyrate was negatively associated with the relative abundances of *Rikenellaceae_RC9_gut_group*, *Prevotellaceae_NK3B31_group*, and *Prevotellaceae_UGG-003*. The molar proportion of propionate was negatively correlated with *Prevotellaceae_NK3B31_group*. Furthermore, the molar proportion of isovalerate was negatively correlated with that of *Prevotellaceae_UGG-003.*

### 3.7. Predicted Function of Rumen Bacteria

To assess the functional characteristics of the rumen microbiota, we used PICRUSt2 to predict potential functions and compared the differences between the two components (Table 4). At KEGG level 2, the relative abundances of amino acid metabolism, carbohydrate metabolism, cell motility, lipid metabolism, and metabolism of terpenoids and polyketides pathways significantly increased in the GYS group (*p* < 0.05). Nevertheless, the pathways of glycan biosynthesis, metabolism, and metabolism of cofactors and vitamins were significantly higher in the GY group than in the GYS group (*p* < 0.05).

## 4. Discussions

There has been much research on the effect of concentrate supplementation on ruminants [27,28]; however, the results have also been different owing to seasonal differences [6,29]. As we expected, the ADG and BW were higher in the GYS group than in the GY group, consistent with other studies on yaks fed with concentrated supplements in summer [29,30]. These results suggest that concentrate supplementation cloud be effective in improving the performance of grazing yaks. Several results have been reported in different breed cattle, in which a significant increase in ADG was observed with an increase in dietary energy or protein level [31,32]. In the present study, the dietary protein and energy levels were increased by supplying a concentrate, which resulted in higher ADG values. These results suggest that warm-season pastures do not meet the nutritional requirements of yaks to reach their maximum growth potential, and supplemental feeding is still required.

The serum glucose concentration reflects energy utilisation of ruminants. When serum glucose concentration decreases, it signifies a lack of dietary energy or poor energy utilisation [33]. There was a positive effect of concentrate supplementation on the serum concentration of GLU as we expected, which has also reported in goats supplemented with concentrate [34]. This was likely due to the concentrate supplementation providing more dietary energy supply. Under normal physiological conditions, an elevation in TP levels in blood is associated with better dietary protein nutrition [35]. As expected, the serum TP concentration, primarily albumin, and globulin, was significantly increased with supplemental feeding, consistent with Jim et al. [36], who investigated the effects of concentrate level on Tan lambs. As for serum AST, the concentration in the GYS group was significantly higher than that in the GY group. In a similar study, Chen et al. [37] reported that the serum concentration of AST increased with the concentration of concentrate in housing-feeding yaks.

Rumen pH is a direct indicator of the stability of the rumen environment, with a normal level of 5.5–7.5 [38]. Several factors affect rumen pH, including salivary secretion, regurgitation time, diet structure, and accumulation of organic acids in the rumen [39,40]. A low pH was reported in yaks that received concentrate supplementation of this study, which is consistent with the results of Wang et al. [41] and Mckay et al. [42]. This may have been due to the supplementation providing more easily degradable carbohydrates and the production of substantial amounts of VFAs by rumen microorganisms. Several studies reported that increased dietary concentrate level could promote growth NH_3_-N and MCP concentrations [43,44], our study obtained consistent results. VFAs are the dominant products of rumen fermentation in ruminants, providing approximately 65–70% of the energy for ruminants. As expected, GYS group showed greater VFA production with a lower acetate-to-propionate ratio. Concentrate intake at 1% LW was enough to promote changes on bacterial population and consequently on rumen fermentation that caused this response. These effects have been reported previously [45,46,47].

In this study, high-throughput sequencing technology was used to investigate the effect of supplemental concentrate feed on the rumen microbiota of yaks during the warm season. In accordance with previous studies on ruminants [11,48,49,50], the current study showed that the bacterial species diversity and richness were significantly higher in the GYS group than in the GY group. Furthermore, beta diversity analysis indicated a visible separation between the samples from the distinct feeding types. Based on the above analysis, there was a notable reduction in the ruminal microbial composition due to supplemental concentrate feeding. This was probably because supplemental concentrate feed with low pH values restrained the growth of some fiber-degrading rumen bacteria, as described in Shen’s study [51]. In this study, we found that Firmicutes had the highest relative abundance, followed by Bacteroidetes, Saccharibacteria, and Fibrobacteres in the yak rumen, similar to previous studies [4,52]. Notably, the relative abundance of Firmicutes was significantly higher in the GYS group than that in the GY group. The relative abundance of Bacteroidetes was significantly higher in the GY group than in the GYS group. This phenomenon was reported by Zhang et al. [50] and Sandra et al. [53].

At the genus level, the relative abundance of the *Christensenellaceae_R-7_group* was significantly higher in the GYS group than in the GY group. *Christensenellace-ae_R-7_group* have been reported as those containing genes for essential cellulase and hemicellulase secretase enzymes that may enhance the ability of yaks to degrade plant cellulose and obtain energy from indigestible polysaccharides [54]. Notably, the *Christensenellaceae_R-7_group* was positively correlated with butyrate, which was linear, with butyrate concentration being significantly higher in the GYS group than in the GY group, consistent with previous studies [4,55]. Previous studies have shown that bacteria from the Ruminococcaceae family play a crucial role in fiber degradation and biohydrogenation [56,57]. Our study revealed that the relative abundance of Ruminococcus_1 was significantly higher in the GY group than in the GYS group, indicating that yaks had a higher ability to degrade fibre in the GY group, which was also reflected by the concentration of acetate, which was significantly higher in the GY group than in the GYS group. Nevertheless, the relative abundance of *Ruminococcaceae_NK4A214_group* was significantly lower in the GY group than in the GYS group, consistent with the results of previous studies where high-concentrate diets were fed to other ruminants [58]. Furthermore, a previous study revealed that the *Ruminococcaceae_NK4A214_group* was positively correlated with isobutyrate and isovalerate concentrations [59]. In the current study, the higher abundance of *Ruminococcaceae_NK4A214_group* in the GYS group might explain the lower butyrate, valerate, isobutyrate, and isovalerate concentrations in the GY group.

Prevotella is the most abundant genus of Bacteroidetes and can reduce nitrogen losses, producing succinic and acetic acids as the primary fermentation end-products of glucose metabolism and improving the utilisation of forage feed types [60]. Thus, it is meaningful that the genera *Prevotellaceae_UCG_003* and *Prevotellace-ae_UCG_005* were higher in the GY group. Previous studies have shown that the *Rikenellaceae_RC9_gut_group* is primarily involved in the degradation of plant-derived polysaccharides, and its relative abundance increases with increasing dietary fibre content [45,61]. In this study, we observed that the *Rikenellaceae_RC9_gut_group* was significantly positively correlated with acetate production. Therefore, supplementary feeding increased the proportion of easily degradable carbohydrates in the diet of the yaks. Succiniclasticum, which belongs to Firmicutes, can produce succinate, convert it into propionate, and further produce glucose [62]. Luo et al. [63] showed that Succiniclasticum concentration in the rumen increased with increasing levels of concentrate, which was inconsistent with the present study results, probably due to difference in the study animals or diet structures.

Microorganisms are essential for an animal’s immunity, nutrient degradation, absorption, and enzyme metabolism [64]. In this study, PICRUSt2 analysis was used to predict the function of the yak rumen microbial community in the different treatment groups. Notably, in KEGG pathway level 2, the relative abundances of amino acid, carbohydrate, and lipid metabolism pathways were significantly increased in the GYS group, consistent with a previous study on the effects of supplemental concentrates on metabolism in grazing Simmental Heifers [27]. Nevertheless, our study investigated the effects of concentrate supplementation on growth performance, serum biochemical parameters, rumen fermentation, and microbial structure of grazing yaks during the warm season. Further studies should investigate appropriate supplementation and its role in protecting the ecological environment.

## 5. Conclusions

Concentrate supplementation (18% CP, 12.49 MJ ME/kg) at the level of 1% in grazing yaks during the warm season increased gain weight as result of improves on ruminal fermentation efficiency mainly by an increase in MCP yield and VFA production and by a lower acetate-to-propionate ratio. Those changes promoted better blood biochemical parameters. Concentrate supplementation decreased bacterial richness, diversity in dices, and relative abundance of *succiniclasticum*, *Prevotellaceae_UCG_003*, *Prevotel-laceae_UCG_005*, and *Ruminococcus_1*, whereas they increased the relative abundances of *Christensenellaceae_R-7_group* and *Ruminococca-ceae_NK4A214_group*. Therefore, this study demonstrated that supplemental feeding of concentrates to grazing yaks is necessary during the warm season in the Qinghai–Tibetan Plateau.

## Figures and Tables

**Figure 1 animals-12-01398-f001:**
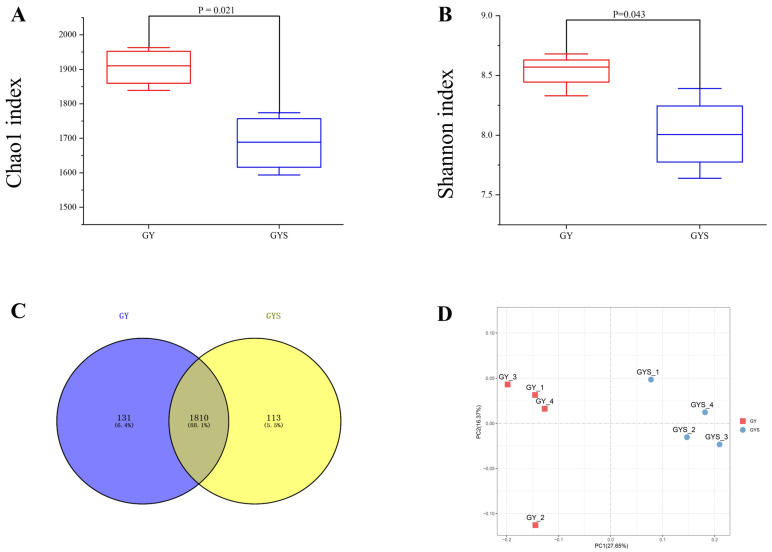
Diversity and richness indicators and OTUs of ruminal bacteria, among different feeding types. GY: grazing group, GYS: concentrate supplement group. (**A**) The Shannon index of different feed types. (**B**) The Chao1 index of different feed types. (**C**) A Venn diagram showing the different and similar OTUs between feeding types. (**D**) Principal coordinate analysis (PCoA) of rumen microbial communities. Data are shown as mean ± SEM.

**Figure 2 animals-12-01398-f002:**
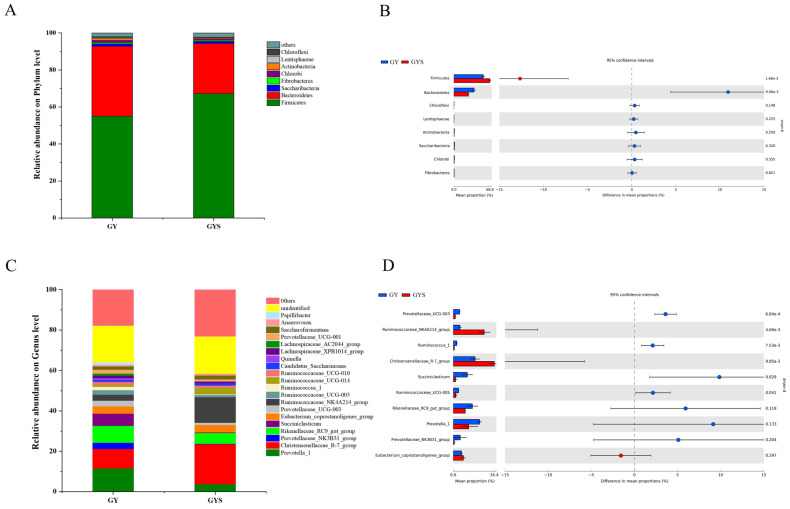
Classification of the bacterial community composition from different feed types. (**A**) Relative abundance of bacterial community at the phylum level. (**B**) Statistical comparison of the dominant phyla in the GY and GYS groups. (**C**) Relative abundance of bacterial community at the genus level. (**D**) Statistical comparison of the dominant genera in the GY and GYS groups. Data are shown as mean ± SEM.

**Figure 3 animals-12-01398-f003:**
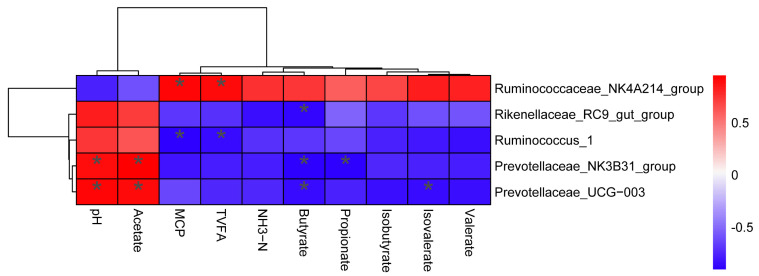
Spearman correlation between the rumen fermentation parameters and predominant genera. * indicates a significant level of 0.05.

**Table 1 animals-12-01398-t001:** Nutrient level of the forage and concentrate supplement (DM%).

Items	Forage	Concentrate Supplement
Nutrient composition, % of DM		
DM ^(1)^	32.51	87.64
Crude protein	11.35	18.76
Crude fat	2.53	2.63
Neutral detergent fiber	54.16	16.03
Acid detergent fiber	26.74	7.83
Rumen degradable protein	-	9.32
Soluble carbohydrates	-	48.56
Calcium	2.15	0.60
Phosphorus	0.08	0.73
ME, MJ/kg ^(2)^	8.66	12.49

^(1)^ DM: Dry matter, ^(2)^ ME: metabolisable energy. ME was calculated according to the Chinese Feed Ingredients and Nutritional Value Table (29th edition).

**Table 2 animals-12-01398-t002:** Effect of concentrate supplementation on body weight and blood indices of grazing yaks during the warm season.

Items ^1^	Group ^2^		SEM ^3^	*p*-Value
	GY	GYS		
Initial body weight (kg)	124.05	124.60	3.293	0.624
Final body weight (kg)	137.00	159.60	4.126	0.041
ADG (g)	215.83	583.33	35.487	0.007
GLU(mmol/L)	4.15	4.54	0.135	0.004
TG (mmol/L)	0.17	0.20	0.012	0.086
TP (mmol/L)	73.17	78.92	1.255	0.029
UN (mmol/L)	5.79	5.45	0.389	0.056
AST(mmol/L)	80.38	94.31	3.241	0.032
ALP (mmol/L)	189.75	206.36	6.125	0.860

^1^ ADG: average daily weight, GLU: serum glucose, TG: triglycerides, TP: total protein, UN: Urea nitrogen, AST: aspartate aminotransferase, ALP: alkaline phosphatase ^2^ GY: grazing group, GYS: concentrate supplement group. ^3^ SEM: standard error of the mean.

**Table 3 animals-12-01398-t003:** Effect of concentrate supplementation on rumen fermentation parameters of grazing yaks during the warm season.

Items ^1^	Group ^2^		SEM ^3^	*p*-Value
	GY	GYS		
pH	7.40	6.86	0.794	<0.001
NH_3_-N, mg/dL	8.10	10.42	0.383	0.001
MCP, mg/L	2.07	2.75	0.116	0.002
TVFA, mmol/L	67.98	80.76	1.252	<0.001
VFA, %				
Acetate	76.44	68.80	0.658	<0.001
Propionate	15.06	19.29	0.487	0.004
Isobutyrate	1.22	1.60	0.094	0.007
Butyrate	5.65	9.31	0.381	0.002
Isovalerate	1.01	1.74	0.012	0.001
Valerate	0.61	1.23	0.117	0.002
A:P	5.07	3.08	0.117	<0.001

^1^ NH3-N: ammonia–N, MCP: microbial protein, TVFA: total volatile fat acid, A:P: acetate acid/propionate acid. ^2^ GY: grazing group, GYS: concentrate supplement group. ^3^ SEM: standard error of the mean.

**Table 4 animals-12-01398-t004:** Effect of concentrate supplementation on predict functions in the bacterial community of grazing yaks during the warm season.

Item	Group ^1^		SEM ^2^	*p*-Value
	GY	GYS		
Amino acid metabolism	13.09	13.28	0.043	0.022
Biosynthesis of other secondary metabolites	2.28	2.06	0.067	0.119
Carbohydrate metabolism	13.50	13.86	0.088	0.040
Cell growth and death	1.62	1.58	0.013	0.056
Cell motility	2.44	3.15	0.139	<0.001
Cellular community prokaryotes	0.18	0.19	0.003	0.047
Digestive system	0.07	0.04	0.005	0.008
Endocrine system	0.25	0.24	0.005	0.299
Energy metabolism	5.75	5.78	0.016	0.860
Environmental adaptation	0.22	0.24	0.005	0.006
Folding, sorting and degradation	3.25	3.27	0.010	0.945
Glycan biosynthesis and metabolism	4.45	3.73	0.147	0.003
Immune system	0.09	0.09	0.001	0.092
Infectious disease	0.18	0.18	0.001	0.093
Lipid metabolism	4.72	5.39	0.137	0.005
Membrane transport	1.52	1.58	0.022	0.236
Metabolism of cofactors and vitamins	13.49	12.63	0.178	0.006
Metabolism of other amino acids	7.11	6.68	0.123	0.112
Metabolism of terpenoids and polyketides	9.73	10.29	0.115	0.003
Nucleotide metabolism	2.12	2.10	0.009	0.288
Replication and repair	6.45	6.46	0.024	0.813
Translation	5.03	5.11	0.022	0.055
Xenobiotics biodegradation and metabolism	2.23	1.81	0.234	0.412

^1^ GY: grazing group, GYS: concentrate supplement group. ^2^ SEM: standard error of the mean.

## Data Availability

The data that support the findings of this study are available from the corresponding author upon reasonable request, and the sequencing data are available from NCBI. The BioProject number is PPRJNA635784.

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
