# Peer review of "Effects of Concentrate Supplementation on Growth Performance, Rumen Fermentation, and Bacterial Community Composition in Grazing Yaks during the Warm Season"

_animals, 2022, doi:10.3390/ani12111398_

Round 1

Reviewer 1 Report

Researchers evaluated the effect of a concentrate supplementation to yaks during the warm season by evaluating various biological aspects.
In literature there are many works on yaks, on microbial communities in yaks and cows, and on dietary supplementation effects. It is a hot topic clearly as the authors themselves say in the text.
It is interesting that they evaluated the effects during the warm period.

I would have analyzed more animals honestly.

Some details on sequencing are missing.
It is important to submit RAW sequences to a public database as NCBI and add the SRA access number in the text for scientific transparency.
Add information about the type of sequencing (length and SE / PE)
How many sequences did you obtain?
Are rarefaction curves okay?
Which normalization and transformation method were used for data analysis?

The conclusions are consistent with the evidence and arguments
presented, and they address the main question posed.

References are fine, up-to-date, and relevant.

Reviewer 2 Report

Line 91: “The dominant species were small, alpine, dwarf, and thread leaf songgrasses”. Include the scientific name of the main forage species evaluated during the experimental period.

Line 104. Table 01. Include the ingredients used to formulate the supplement.

Line 107 to 109: “Yak weight was measured continuously at the beginning and end of the trial using 107 floor scales, and the average daily weight gain (ADG) was calculated as = (Final BW- Final 108 BW)/60(g/day)”. Justify why the animals were not fasted before weighing at the beginning and end of the experimental period.

Line 132-134: “The neutral detergent fibre (NDF) and acid detergent fibre (ADF) contents were determined using methods described by Van Soest et al. [22]”. Justify why you used alpha amylase to evaluate the forage fibrous fraction.

Line 167: “Data and Statistical Analysis”. Were samplings and weighing carried out during the 60 days of collection data? Few replications were used to evaluate daily weight gain, we suggest including intermediate weights in the statistical analyses.

Line 291. Discussions. All the supplement supplied were consumed by the animals? Considering the daily weight gains of the treatments GY (215.83 g/day) and GYS (583.33 g/day), what are the total CP and energy requirements to meet these gains? What were the amounts of nutrients consumed from the supplement?

Author Response

Response to Reviewer 2Comments

Dear reviewer:

We are very grateful for your efforts on our manuscript and for giving us the opportunity to resubmit a revised version of our manuscript. Those comments are all valuable and very helpful for revising and improving our paper, as well as the important guiding significance to our researches. We have studied comments carefully and have made correction which we hope meet with approval. Revised portion are marked in red in the paper. The main corrections in the paper and the responds to the reviewer’s comments are as flowing:

Point 1: The dominant species were small, alpine, dwarf, and thread leaf songgrasses”. Include the scientific name of the main forage species evaluated during the experimental period.

Response 1: According to the comments: we have corrected: The dominant species of the forage were Elymus nutansv, Kobresia humili, Poa pratensiv, Potentill abifurca, Saussurea pulchra, and Ajania tenuifolia during the experimental period. (L95-96, page 3).

Point 2: Yak weight was measured continuously at the beginning and end of the trial using floor scales, and the average daily weight gain (ADG) was calculated as = (Final BW- Final 108 BW)/60(g/day)”. Justify why the animals were not fasted before weighing at the beginning and end of the experimental period.

Response2: We are so about this problem that shouldn't occur, owing to the general idea of writing and not describing clearly, have been revised and added “The body weight was weighed at the beginning and end of the trial using floor scales before the morning feeding” to “Yak weight was measured continuously at the beginning and end of the trial using floor scales”.(L114-115, page 4).

Point 3: The neutral detergent fibre (NDF) and acid detergent fibre (ADF) contents were determined using methods described by Van Soest et al. [22]”. Justify why you used alpha amylase to evaluate the forage fibrous fraction.

Response3: We are grateful for the comments: We thought about it this way, because the feed sample is a mixture, not a fixed chemical entity, in the process of determining crude fiber part of the cellulose, lignin, hemicellulose dissolved in sodium hydroxide solution is excluded. This is lower than the actual value, and in the determination of the index we have consulted relevant references that also follow this method by Van Soest et al. References such as:

  1. Ahmad AA, Zhang JB, Liang Z, et al. Dynamics of rumen bacterial composition of yak (Bos grunniens) in response to dietary supplements during the cold season. PeerJ. 2021, 9:e11520.
  2. Liu H, Jiang H, Hao L, et al. Rumen Bacterial Community of Grazing Lactating Yaks (Poephagus grunniens) Supplemented with Concentrate Feed and/or Rumen-Protected Lysine and Methionine. Animals (Basel). 2021;11(8):2425.

Point 4: “Data and Statistical Analysis”. Were samplings and weighing carried out during the 60 days of collection data? Few replications were used to evaluate daily weight gain, we suggest including intermediate weights in the statistical analyses.

Response4: Thank your suggestion, unfortunately, we did not weigh the yaks in the middle of the trial. Since yaks are more difficult to weigh compared to other animals, and of course mainly due to our negligence when doing the experiment. Furthermore, the relevant papers did as the statistical analyses.

  1. Xue BC, Zhang JX, Wang ZS, et al. Metabolism response of grazing yak to dietary concentrate supplementation in warm season. Animal. 2021;15(3):100175.

Point 5: Discussions. All the supplement supplied were consumed by the animals? Considering the daily weight gains of the treatments GY (215.83 g/day) and GYS (583.33 g/day), what are the total CP and energy requirements to meet these gains? What were the amounts of nutrients consumed from the supplement?

Response5: We are grateful for the suggestion: We have made the following statements on the above those problems.

  1. The yaks in the supplementation group received 1 % of daily concentrate supplementation body weight per yakin this experiment, we observed the entire trial period, all the supplement supplied were consumed by the animals.
  2. Based on the references (1,2), we calculate the treatments GY of the total CP and energy requirements were 230.40 g/d and 35.84 MJ/d, and treatments GYS were 304.45 g/d and 48.07 MJ/d.
  3. We are extremely grateful to reviewer for pointing out this problem, while the pasture feed intake was difficult to measure due to the circumstances of the study. Therefore, it was difficult to calculate the amounts of nutrients consumed from the supplement, and we sincerely regret this. Some similar studies (3,4) also did not measure feed intake, such as:
  4. Xue B, Chai S T, Liu S J, et al. Protein requirements of yaks during the growth period. Chinese Qinghai Journal of Animal and Veterinary Sciences.1994, 4, 41-45.
  5. Hu LH, Xie A Y, Han X T, et al. Study of energy metabolism and rumen metabolism in cattle consumed during the growth period. Chinese Qinghai Journal of Animal and Veterinary Sciences. 1992, 4, 1-6.
  6. Xue BC, Zhang JX, Wang ZS, et al. Metabolism response of grazing yak to dietary concentrate supplementation in warm season. Animal. 2021;15(3):100175.
  7. Xie R, Zheng Q, Luo  Finishing effect of Maiwa yak by supply feed in warm season Grass-feeding. Livestock, 2004, 4 , 56-58

Our deepest gratitude goes to you for your careful work and thoughtful suggestions that have helped improve this paper substantially.

Round 2

Reviewer 2 Report

Animals-1725966-Effects of Concentrate Supplementation on Growth Performance, Rumen Fermentation, and Bacterial Community Composition in Grazing Yaks during the Warm Season

Line 115 to 117: “Yak weight was measured continuously at the beginning and end of the trial using floor scales, and the average daily weight gain (ADG) was calculated as = (Final BW- Final BW)/60(g/day)”. Justify why the animals were not fasted before weighing at the beginning and end of the experimental period.

Line 142-143: “The neutral detergent fibre (NDF) and acid detergent fibre (ADF) contents were determined using methods described by Van Soest et al. [22]”. Justify why you used alpha amylase to evaluate the forage fibrous fraction.

Line 180: “Data and Statistical Analysis”. Were samplings and weighing carried out during the 60 days of collection data? Few replications were used to evaluate daily weight gain, we suggest including intermediate weights in the statistical analyses.

Line 310-315: ” These results suggest that concentrate supplementation cloud be effective in improving the performance of grazing yaks. Many results have been reported in different breed cattle, in which a significant increase in ADG was observed with an increase in dietary energy or protein level [31,32]. In the present study, the dietary protein and energy levels were increased by supplying a concentrate, which 314 resulted in higher ADG values”. What are the total CP and energy requirements to meet these gains? What were the amounts of nutrients consumed from the supplement?

Author Response

Dear reviewer:

We are very grateful for your efforts on our manuscript and for giving us the opportunity to resubmit a revised version of our manuscript. Those comments are all valuable and very helpful for revising and improving our paper, as well as the important guiding significance to our researches. We have studied comments carefully and have made correction which we hope meet with approval. Revised portion are marked in blue in the paper. The main corrections in the paper and the responds to the reviewer’s comments are as flowing:

Point 1: Line 115 to 117: “Yak weight was measured continuously at the beginning and end of the trial using floor scales, and the average daily weight gain (ADG) was calculated as = (Final BW- Final BW)/60(g/day)”. Justify why the animals were not fasted before weighing at the beginning and end of the experimental period.

Response2: Due to the lack of seriousness in writing, in reality, we used fasting before weighing the yak, we have carefully revised the manuscript.(L114, page 3).

Point 2: Line 142-143: “The neutral detergent fibre (NDF) and acid detergent fibre (ADF) contents were determined using methods described by Van Soest et al. [22]”. Justify why you used alpha amylase to evaluate the forage fibrous fraction.

Response2: We are grateful for the comments: Warm season forage and concentrate supplement contained a certain amount of starch and if we did not use alpha amylase to evaluate the fiber portion of the feed, some starch will not dissolved thus increasing the fiber content. Relevant references that also follow this method by Van Soest et al. References such as:

  1. Ahmad AA, Zhang JB, Liang Z, et al. Dynamics of rumen bacterial composition of yak (Bos grunniens) in response to dietary supplements during the cold season. PeerJ. 2021, 9:e11520.
  2. Liu H, Jiang H, Hao L, et al. Rumen Bacterial Community of Grazing Lactating Yaks (Poephagus grunniens) Supplemented with Concentrate Feed and/or Rumen-Protected Lysine and Methionine. Animals (Basel). 2021;11(8):2425.

Point 3: Line 180: “Data and Statistical Analysis”. Were samplings and weighing carried out during the 60 days of collection data? Few replications were used to evaluate daily weight gain, we suggest including intermediate weights in the statistical analyses.

Response3: Thank your suggestion, unfortunately, we did not weigh the yaks in the middle of the trial. Furthermore, the main objective of this manuscript was to investigate the changes in growth performance, serum biochemistry and microbiology at the beginning and end of supplementation.  Furthermore, the relevant papers did as the statistical analyses.

  1. Xue BC, Zhang JX, Wang ZS, et al. Metabolism response of grazing yak to dietary concentrate supplementation in warm season. Animal. 2021;15(3):100175.

Point 4: Line 310-315: These results suggest that concentrate supplementation cloud be effective in improving the performance of grazing yaks. Many results have been reported in different breed cattle, in which a significant increase in ADG was observed with an increase in dietary energy or protein level [31,32]. In the present study, the dietary protein and energy levels were increased by supplying a concentrate, which 314 resulted in higher ADG values”. What are the total CP and energy requirements to meet these gains? What were the amounts of nutrients consumed from the supplement?

Response5: We are grateful for the suggestion: we calculate the treatments GY of the total CP and energy requirements were 230.40 g/d and 35.84 MJ/d, and treatments GYS were 304.45 g/d and 48.07 MJ/d, while the pasture feed intake was difficult to measure due to the circumstances of the study. Therefore, it was difficult to calculate the amounts of nutrients consumed from the supplement, it is limitation of this article, and we sincerely regret this. Some similar studies (1,2) also did not measure feed intake, such as:

  1. Xue BC, Zhang JX, Wang ZS, et al. Metabolism response of grazing yak to dietary concentrate supplementation in warm season. Animal. 2021;15(3):100175.
  2. Xie R, Zheng Q, Luo  Finishing effect of Maiwa yak by supply feed in warm season Grass-feeding. Livestock, 2004, 4 , 56-58.

Point 5: English language and style need moderate English changes required.

Point 5: We requested a professional agency to do the language touch-ups, and this time it has been improved. Altered areas are marked in blue in the text.

Our deepest gratitude goes to you for your careful work and thoughtful suggestions that have helped improve this paper substantially.
